# Calcium- and Voltage-Dependent Dual Gating ANO1 is an Intrinsic Determinant of Repolarization in Rod Bipolar Cells of the Mouse Retina

**DOI:** 10.3390/cells9030543

**Published:** 2020-02-26

**Authors:** Sun-Sook Paik, Yong Soo Park, In-Beom Kim

**Affiliations:** 1Department of Anatomy, College of Medicine, The Catholic University of Korea, Seoul 100744, Korea; paikss@catholic.ac.kr (S.-S.P.); pakoshahaha@catholic.ac.kr (Y.S.P.); 2Catholic Institute for Applied Anatomy, College of Medicine, The Catholic University of Korea, Seoul 100744, Korea

**Keywords:** anoctamin 1, TMEM16A, rod bipolar cell, retina, repolarization, patch-clamp

## Abstract

TMEM16A/anoctamin1 (ANO1), a calcium (Ca^2+^)-activated chloride (Cl^−^) channel, has many functions in various excitable cells and modulates excitability in both Ca^2+^- and voltage-gating modes. However, its gating characteristics and role in primary neural cells remain unclear. Here, we characterized its Ca^2+^- and voltage-dependent components in rod bipolar cells using dissociated and slice preparations of the mouse retina. The I-V curves of Ca^2+^-dependent ANO1 tail current and voltage-gated Ca^2+^ channel (VGCC) are similar; as ANO1 is blocked by VGCC inhibitors, ANO1 may be gated by Ca^2+^ influx through VGCC. The voltage-dependent component of ANO1 has outward rectifying and sustained characteristics and is clearly isolated by the inhibitory effect of Cl^−^ reduction and T16Ainh-A01, a selective ANO1 inhibitor, in high EGTA, a Ca^2+^ chelator. The voltage-dependent component disappears due to VGCC inhibition, suggesting that Ca^2+^ is the essential trigger for ANO1. In perforated current-clamping method, the application of T16Ainh-A01 and reduction of Cl^−^ extended excitation periods in rod bipolar cells, revealing that ANO1 induces repolarization during excitation. Overall, ANO1 opens by VGCC activation during physiological excitation of the rod bipolar cell and has a voltage-dependent component. These two gating-modes concurrently provide the intrinsic characteristics of the membrane potential in rod bipolar cells.

## 1. Introduction

The retina is a specialized tissue that receives and converts the light into neural signals that travel to the central nervous system (CNS) where visual information, such as shape, color, contrast, etc., is processed and encoded. To perform this complex processing, the retina forms and uses multiple parallel pathways [1,2,3]. The retinal bipolar cells are a second-order neuron that transfers visual information from photoreceptors to amacrine and ganglion cells and is, therefore, a key component of parallel visual processing. It is a unique non-spiking neuron that uses graded changes in membrane potential [4,5,6]. However, little is known about the physiology of bipolar cells and its mechanism.

Calcium (Ca^2+^)-activated chloride (Cl^−^) channels (CaCCs) have been identified in various cells where they play manifold roles in cell physiology [7,8,9,10,11,12]. In the retina, multiple types of CaCCs, including CLCA, bestrophin, CLC families, TMEM16A/anoctamin 1 (ANO1), and TMEM16B (ANO2) are expressed [13,14,15]. ANO1 and ANO2 are mainly found in photoreceptor and bipolar cell terminals [13,15,16], whereas CLCA and bestrophin are found in retinal pigment epithelium and Müller cells [14]. Calcium-activated chloride currents (I_Cl(Ca)_) have been described in salamander and goldfish retinas. I_Cl(Ca)_ is generally activated by Ca^2+^-influx and maintains the balance of the feedback response from the horizontal cells to photoreceptors [17,18,19]. Its possible role in bipolar cells has been reported in dissociated goldfish and mouse retinas [13,20]. However, the characteristics and functional significance of CaCC in bipolar cells remain unknown.

The expressions and functions of ANO1 and ANO2 have been identified in various sensory neurons such as olfaction [21,22,23], vision [13,16,24], hearing [25,26], taste [27], and pain and temperature [28]. Recently, ANO2 in central neurons, such as hippocampal pyramidal neurons [29], thalamocortical neurons [30], inferior olive neurons [31], and cerebellar Purkinje cells [32,33], has been anatomically and functionally characterized. However, although it is known that ANO1 exhibits voltage-gating mode, the functional role of ANO1 in central neurons remains to be elucidated. A few studies reported ANO1 expression in the retina [13], auditory brainstem [34], and cerebellum [33,35]. In addition, the function and gating mechanism of a distinct feature of ANO1 has been partially explained in transfected cell lines in vitro [36,37,38,39]; however, the specific role played by ANO1 in primary CNS cells remains unknown.

Therefore, this study aimed to determine the gating mechanism of endogenous ANO1 in mouse retinal rod bipolar cells, which strongly express ANO1 and have I_Cl(Ca)_ [13]. Further, we seek to explain how it contributes to neural cell excitation through its endogenous property. Using patch-clamp recordings, we identified that the physiological activity of ANO1 in rod bipolar cells is closely related to the Ca^2+^ channel in retinal slices. We also isolated the voltage-dependent ANO1 current through the step-voltage protocol. Lastly, we demonstrated that Ca^2+^ and voltage dual gating modes operate in rod bipolar cells and modulate their excitation.

## 2. Materials and Methods

### 2.1. Ethical Standards

All animal experiments followed the regulations of the Catholic Ethics Committee of The Catholic University of Korea, Seoul, which are based on the guidelines of the National Institute of Health (NIH) for the Care and Use of Laboratory Animals (NIH Publications No. 80-23) revised in 1996. The protocol was approved by the IACUC (Institutional Animal Care and Use Committee) of the College of Medicine, The Catholic University of Korea (Approval Number: CUMC 2017-0241-11 and 2019-0079-04).

### 2.2. Preparation of Animals and Tissue Samples

C57BL/6 mice, 7–8 weeks of age, were used. The mice were kept in 12 h light/dark cycle in a climate-controlled laboratory. Before retinal dissection, the mice were anesthetized with zolazepam (20 mg/kg) and xylazine (7.5 mg/kg) intraperitoneally and sacrificed by high dose injection of zolazepam after eye enucleation. The cornea and lens were immediately separated from the eye, and the retina was gently dissected from the sclera and placed in iced extracellular solution.

For the rod bipolar cell dissociation, the retina was incubated in low Ca^2+^ solution with 4 mg/mL of papain. The cells were enzymatically isolated from the retina and kept in an oxygenated artificial extracellular solution. The dissociated rod bipolar cells were identified based on shape, having spheroid soma and stout axon with large lobulated terminals [13,40].

For the retinal slice preparation, the retina was dissected in iced bicarbonate buffer saline, oxygenated by 95% O_2_–5% CO_2_ gas and containing (in mM): NaCl 126, KCl 2.5, CaCl_2_ 2.4, MgCl_2_ 1.2, NaH_2_PO_4_ 1.2, NaHCO_3_ 18, and glucose 11. The retina was then embedded in 1.5% agar solution and sectioned into 200 μm-thick samples using vibratome (Campden instrument, Loughborough, England). They were incubated in the oxygenated extracellular solution at room temperature (24–27 ℃). The cell bodies on the top of the inner nuclear layer or just below the outer plexiform layer were selected as rod bipolar cells. Alexa 488 dye was diluted with the internal solution to visualize the axon terminals, which are located in the inner portion of the inner plexiform layer.

### 2.3. Preparation of Solutions and Drugs

The extracellular solution was continuously perfused to the bath with 95% O_2_–5% CO_2_ and containing (in mM): NaCl 130, CsCl 5, CaCl_2_ 2.5 MgCl_2_ 1, glucose 10, and HEPES 10. The pH was adjusted to 7.4 with NaOH. To change its Cl^−^ concentration, NaCl was replaced with Na-gluconate. The pipette solution for the voltage-clamp method contained (in mM): CsCl 140, MgCl_2_ 1, CaCl_2_ 0.5, and Tris-ATP 10. Its pH was adjusted to 7.2 with CsOH. The calculated Cl^−^ reversal potential was 0 mV. EGTA or BAPTA were added to the pipette solution to reduce Ca^2+^ signaling. The pipette solution for the perforated current-clamp method contained (in mM): KCl 140, MgCl_2_ 1, CaCl_2_ 0.5, EGTA 5, Tris-ATP 10, and 14.8 μg/mL gramicidin D. Its pH was adjusted to 7.2 with KOH.

T16Ainh-A01 and CaCCinh-A01 were used to inhibit I_Cl(Ca)_. Mibefradil was used to inhibit the T-type calcium current; nifedipine, L-type calcium current. All drugs were dissolved in the extracellular solution with 0.1% DMSO.

### 2.4. Data Analysis

The current and voltage signals from the patch-clamp amplifier (EPC-9; HEKA Elektronik, Germany) were monitored and recorded in a conventional manner. The stimulation and acquisition of signals were carried out using the Pulse+PULSE-fit program (HEKA). The signals were filtered at 2 kHz and digitized at 5 kHz with a data acquisition interface (LIH 1600 A/D board; HEKA). The cell membrane capacitance and series resistance current were automatically compensated by the amplifier. The calculated liquid junction potential was also subtracted. Data were analyzed using PULSE-fit and ORIGIN programs. Results were statistically analyzed using Student’s *t*-tests and ANOVA. Tukey’s multiple comparison test was used for post hoc test after ANOVA. All statistical analysis was performed through the Prism 8.0 software. Data are presented as mean ± S.E.M. with the significance set to *p* < 0.05 (*).

## 3. Results

### 3.1. Relationship Between Ca^2+^-Dependent Characteristics of the ANO1 Current and VGCC

Previously, we have demonstrated the Ca^2+^-dependence of ANO1 tail current (I_tail_) in dissociated rod bipolar cells of the mouse retina by increasing [Ca^2+^]_o_ to 10 mM and lowering [EGTA]_i_ to the range of 0–0.5 mM [13]. However, to maximize I_tail_, the values used for the stimulating potential (10 mV) and the concentrations of the Ca^2+^ (10 mM) and EGTA (0.5 mM) were made up conditions. Therefore, in this study, we examined the induction of I_tail_ in conditions that mimic cellular environments using basal [Ca^2+^]_o_ (2.5 mM) and EGTA (1 mM) in retinal slices. From Figure 1A, I_tail_ slowly declined inward at the end of the stimulation and was effectively inhibited by two common ANO1 inhibitors, namely T16Ainh-A01 (40 μM) and CaCCinh-A01 (40 μM) [41,42], followed by time course of the bath application (*n* = 7, *p* < 0.05, Figure 1A). I_tail_ was reduced by 5 mM BAPTA (*n* = 7, *p* < 0.05, Figure 1B). These suggest that I_tail_ is mediated by ANO1 and is activated by Ca^2+^.

To determine when ANO1 showed maximal response, we stimulated the rod bipolar cells from −60 to 20 mV with a 10-mV interval and −70 mV as the holding potential. The current traces and the normalized current area of the I_tail_ are presented in Figure 1C (*n* = 7). Interestingly, I_tail_ started to appear at −40 mV and was maximized at the range between −30 and −20 mV. The voltage profile of I_tail_ was similar to that of VGCC elucidated previously using dissociated rod bipolar cells [43,44]. To determine the relationship between I_tail_ and VGCC, we perfused 40 μM mibefradil and 40 μM nifedipine, which are T-type and L-type VGCC inhibitors, respectively. I_tail_ was successfully inhibited by both and was almost completely inhibited upon their simultaneous application (*n* = 7, *p* < 0.05, Figure 1D).

### 3.2. Isolation of the Voltage-Dependent and Outward Component of the ANO1 Current

We accidentally found out that I_tail_ in some rod bipolar cells continuously increased when membrane potential is increased, provided that Ca^2+^-current (I_Ca_) is not elicited (*n* = 10/25, Appendix A). I_tail_ started to increase from −30 mV, wherein the maximal response was achieved in Figure 1C, and continuously increased by voltage stimulation. From this, we hypothesized the existence of a voltage-dependent component of the ANO1 current as reported in transfected cell lines [36,37,38,39].

To identify this, we added 5 mM EGTA to reduce Ca^2+^-dependency in dissociated rod bipolar cells because Ca^2+^-induced ANO1 current could mask the voltage-dependent component. After the reduction in I_tail_, we stimulated the cell by applying a voltage from −30 to 20 mV at 10-mV intervals. From the traces in Figure 2A, there is a rectified pattern of outward current in rod bipolar cells. This was reduced in low extracellular Cl^−^ (5 mM) solution, suggesting that a Cl^−^ component exists in the outward current. The amplitude was measured after stimulation in each step and was plotted in Figure 2B. The difference between the amplitude and Cl^−^ concentration was not significant at low voltage, whereas it was significantly amplified by the increase in membrane potential (*n* = 7, *p* < 0.05, Figure 2B). To confirm that the Cl^−^ component is mediated by ANO1, we applied 10 μM mibefradil, 30 μM nifedipine, and 10 μM T16Ainh-A01. The family of the current traces with inhibitors is shown in Figure 2C. The amplitudes measured at 10 and 200 ms after the onset of the depolarizing pulse were plotted against the membrane potential in Figure 2D,E, respectively (*n* = 6). The inhibitory effect was most prominent in T16Ainh-A01, and the two types of VGCC inhibitors also both reduced the Cl^−^ current. The current difference between control and each inhibitory state was widened at 200 ms after the pulse onset, compared with that at 10 ms after the pulse onset, which confirms that the outward rectifying current was mediated by ANO1 (Figure 2D,E).

### 3.3. Involvement of the Voltage-Dependent Component of ANO1 in the Rapid Decay of the Initial Phase of Waveform

To investigate the physiological function of the voltage-dependent component of ANO1 in normal extracellular Ca^2+^ concentration, we first performed perforating patch-clamp in high EGTA (5 mM) intracellular solution to minimize Ca^2+^-dependency and maintain the physiological intracellular Cl^−^ proportion (Figure 3A). After confirming the cells lacking I_tail_ from four consecutive control recordings (boxed gray waveforms), the extracellular solutions were changed in this order: low Cl^−^, control, and with T16Ainh-A01. In each state, we confirmed that the outward current was maximized in the control solution and was decreased by Cl^−^ reduction or ANO1 inhibition (Figure 3A).

Further, we performed the current-clamp method using the same extracellular solutions in the same cells. We stimulated the cells by injecting the depolarizing current from a resting potential of −50 mV and yielded a distinctive waveform with a rapid rising single spike in the initial phase followed by plateau potential (Figure 3B). When voltage-dependent ANO1 was inhibited by reducing [Cl^−^]_o_ from 145 to 5 mM, the decay after a spike to the plateau potential became slower than that of the control solution, and the plateau potential thus disappeared. When the extracellular solution was replaced with the control, the waveform reverted to its distinctive form. ANO1 inhibition showed similar results with Cl^−^ reduction (*n* = 10; Figure 3B).

For the quantitative analysis of the waveforms in Figure 3A,B, we superimposed them and compared the time constant τ of the decay phase after single spike with the amplitude of the spike in each extracellular condition (Figure 3C). τ was obtained from the single exponential fit of the initial phase of waveform, and it was represented in the bar graph with its spike amplitude. Decay time constant was significantly increased by both ANO1 inhibitions via Cl^−^ reduction and T16Ainh-A01 (*p* < 0.05), whereas spike amplitude was not affected (*p* > 0.05). These results indicate that the voltage-dependent component of ANO1 exists in mouse retinal rod bipolar cell and accelerates the repolarization of the membrane potential without changing its amplitude.

## 4. Discussion

Applying the patch-clamp method, we discovered that ANO1 physiological activity was strongly associated with VGCC and that, in rod bipolar cells of the mouse retina, ANO1 showed a voltage-dependent behavior. We also demonstrated that both Ca^2+^- and voltage-dependent gating-modes operated during the excitation of rod bipolar cells, and these two modes accelerated repolarization in the current-clamp mode. Regarding ANO1-specific inhibitors, it is worth mentioning that both CaCCinh-A01 and T16Ainh-A01 reportedly do not pharmacologically discriminate between ANO1 and ANO2 [42]. Nevertheless, CaCCinh-A01 and T16Ainh-A01 have been widely used as ANO1-specific inhibitors [41,42,45]. Moreover, previous immunohistochemical studies demonstrated that the mouse retinal rod bipolar cells did not express ANO2 [16,46], while they strongly expressed ANO1 [13]. Therefore, it is reasonable that, in this study, both inhibitors were used as ANO1-specific inhibitors.

Previously, we reported the ANO1 expression in rod bipolar cells of mouse retina and identified ANO1-mediating I_Cl(Ca)_ using immunohistochemistry and the patch-clamp method [13]. We focused on the identification of ANO1-mediating I_Cl(Ca)_ in primary central neurons. However, the channel activity was not observed under experimental physiological conditions, and the gating mechanism of its distinct feature, which has voltage- and Ca^2+^-gating-modes as elucidated in transfected cell lines, was not determined [36,37,38,39]. Thus, in the present study, we investigated whether ANO1 in rod bipolar cells could open in the physiological excitatory conditions and isolated its voltage-dependent component. In the normal range of extracellular Ca^2+^ concentration with 1 mM EGTA, I_tail_ was apparently evoked by cell stimulation, and it was inhibited by 40 μM each of CaCCinh-A01 and T16Ainh-A01. As Ca^2+^ signaling is important for the excitation and synaptic transmission in mammalian rod bipolar cells [47,48], we hypothesized that ANO1 could follow VGCC activity. Consistent with our hypothesis, the I-V curves of I_tail_ in Figure 1C were similar to those of VGCC, wherein the amplitude is maximized approximately at −20 mV in rod bipolar cells [43,44,47].

Previous studies reported that ANO1 is localized close to L-type VGCC in the photoreceptors [13,15,49]. Previous authors further claimed that BAPTA, a Ca^2+^ chelator, and nifedipine, an L-type VGCC inhibitor, significantly inhibited ANO1 currents in photoreceptors [13,15,49] and rod bipolar cells [13,15,49], suggesting that L-type VGCC is a likely Ca^2+^ source for ANO1. However, because T-type VGCC also contributes to the synaptic transmission in rod bipolar cells [50], we hypothesized that Ca^2+^ via T-type VGCC could affect the gating mechanism of ANO1. Here, mibefradil, a T-type VGCC inhibitor, effectively reduced ANO1 current (Figure 1D, Figure 2C). These results suggest that ANO1 could be activated by any VGCCs, which leads to an increase in intracellular Ca^2+^.

Although the contribution of Ca^2+^ to ANO1 has been studied in various transfected and primary cells [51,52,53], there is little evidence of the presence of its voltage-dependent component in primary neural cells. Therefore, we minimized Ca^2+^ signaling in rod bipolar cells using intracellular EGTA [54]. In this condition, notwithstanding I_tail_ reduction, the outward rectifying current was isolated upon high voltage stimulation (Figure 2). The fast component of the outward rectifying current was less sensitive to the membrane potential, whereas the sustained rectifying component showed voltage dependency (Figure 2D,E). This observation is consistent with the specific features of ANO1 in transfected cell lines [37]. The sustained rectifying component was almost abolished by the application of L-type and T-type VGCC inhibitors, mibefradil and nifedipine, respectively (Appendix A). Overall, both Ca^2+^ and voltage can gate the ANO1 channel. Besides, the initial gating factor is increased by intracellular Ca^2+^ through VGCC, and afterward, membrane voltage sustains ANO1 opening. Our findings are corroborated by previous studies in ANO1-transfected HEK cells [36,39], which demonstrated that intracellular Ca^2+^ gates ANO1 similar to that in a ligand-gated channel and that the voltage-dependent mode is not activated without Ca^2+^.

To investigate the cellular function of ANO1 in rod bipolar cells of the retina, we examined its role during cell excitation using the current-clamp method. The peak amplitude of the single spike was neither affected by Cl^−^ reduction nor ANO1 inhibition, whereas the decay after a spike to the plateau potential was significantly affected, and plateau potential thus disappeared. These results suggest that ANO1 contributes to the rapid repolarization process. It is consistent with previous studies about modulation of the action potential by ANO1 or ANO2, which showed elongation of the excitation by Cl^−^ inhibition [29,55].

During neural excitation, the balance between excitatory and inhibitory signals is essential for proper synaptic transmission. In rod bipolar cells, Ca^2+^ has a crucial role in the excitation and synaptic transmission in the mammalian retina [50,56,57]. However, the mechanism of the intrinsic modulation of the excitation in rod bipolar cells remains unclear. Large-conductance calcium-activated potassium channel (BK) and small-conductance calcium-activated potassium channel (SK) are representative intrinsic modulators of the Ca^2+^-induced excitation [58,59]. However, the BK channel is not expressed in rod bipolar cells, whereas it is localized in the dendrites of the A17 cell [60,61]. Although the SK channel is expressed in horizontal cells, dopaminergic cells, and ganglion cells, it is absent in rod bipolar cells [62]. We found that BK and SK channel inhibitors, iberiotoxin and apamin, did not affect the membrane potential during excitation in rod bipolar cells (Appendix A), confirming the absence of both BK and SK channels. Therefore, as an intrinsic inhibitory modulator of the rod bipolar cell excitation, we conclude that ANO1 could replace the BK and SK channels for regulating cell excitation.

## 5. Conclusions

The present study elucidated the dual gating mode of ANO1 in primary CNS neurons. Although the retinal rod bipolar cell, being a non-spiking neuron, is not typical, it uses graded changes in the membrane potential and contributes to repolarization during cell excitation. As illustrated in Figure 4, an increase in intracellular Ca^2+^ through VGCC rapidly opens ANO1. The ANO1 current is sustained by the synergistic effects of the membrane potential and Ca^2+^, thereby accelerating cell repolarization. These results suggest that ANO1 acts as a rapid intrinsic inhibitory modulator during excitation in rod bipolar cells, similar to BK and SK channels in other neural cells.

## Figures and Tables

**Figure 1 cells-09-00543-f001:**
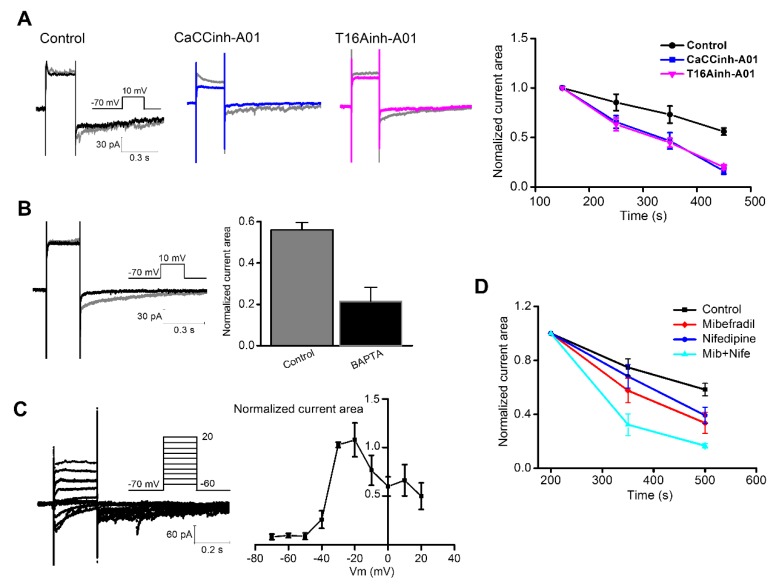
Relationship between Ca^2+^-dependent characteristics of the TMEM16A/anoctamin1 (ANO1) current and voltage-gated Ca^2+^ channel (VGCC). (**A**) Rod bipolar cells were stimulated from a holding potential of −70 mV to a membrane potential 10 mV for 250 ms. Representative current traces before (gray) and 300 s after drug administration (black, blue, pink). The current area was normalized by the area of I_tail_ at 150 s, and the normalized current area over time (right) showed the inhibitory effect of ANO1-specific blockers (*n* = 7; *p* < 0.05, Student’s *t*-test). (**B**) Internal application of 5 mM BAPTA (black; 0.56 ± 0.03) decreased I_tail_ compared with the control (gray; 0.21 ± 0.06; *n* = 7; *p* < 0.05, Student’s *t*-test). (**C**) Rod bipolar cells were stimulated from a holding potential of −70 to −60 mV until approximately 20 mV. Current traces (left) and the normalized current area of I_tail_ (right) against command voltage were plotted. (**D**) The current area was normalized by the area of I_tail_ at 200 s, and the normalized current area over time showed the inhibitory effects of L- and T-type VGCC inhibitors, namely 40 μM nifedipine and 40 μM mibefradil, respectively (*n* = 7; ANOVA, *p* < 0.05).

**Figure 2 cells-09-00543-f002:**
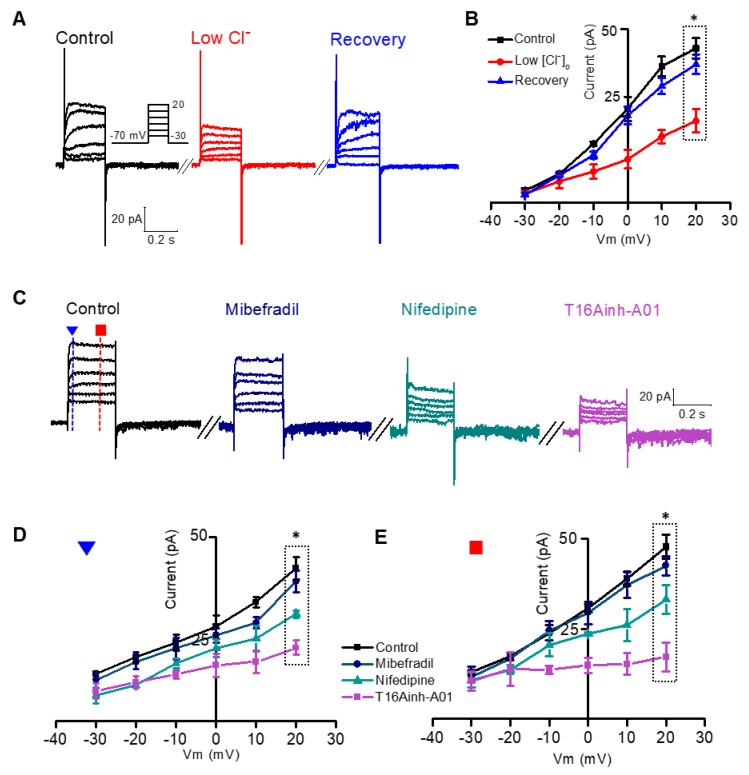
Isolation of the voltage-dependent component of ANO1 in rod bipolar cells. Isolation of the outwardly rectifying ANO1 component with strong voltage dependence. (**A**) In a single rod bipolar cell lacking I_tail_, a family of whole-cell currents was serially recorded to monitor the changes in outward current as [Cl^−^]_o_ changes. First, voltage-clamp (−30 to 20 mV for 250 ms) was performed under 140 mM of [Cl^−^]_o_ (black). Next, a family of whole-cell current was recorded after the extracellular replacement of Cl^−^ with 100 mM gluconate (red) and then back to the control (blue). (**B**) For each of the three cases, the outward current (mean amplitudes measured at 200 ms after the onset of the depolarizing pulse) was plotted against the command voltage (*n* = 7, **p* < 0.05, ANOVA). (**C**) The effects of VGCC inhibitors and ANO1-specific blocker on outward rectifying currents were plotted by recording a series of whole-cell currents. The control (black), 10 μM mibefradil (navy), 30 μM nifedipine (dark cyan), and 10 μM T16Ainh-A01 (violet) were administered in this order to the cells in bath while waiting for recovery before administration. To compare the changes in the earlier portion (blue triangle) and in the later portion (red rectangle), the amplitudes measured at 20 and 200 ms after the onset of the depolarizing pulse were plotted in (**D**) and (**E**) against the command voltage, respectively. The inhibitors significantly reduced outward currents, compared to the control (*n* = 6, **p* < 0.05, ANOVA).

**Figure 3 cells-09-00543-f003:**
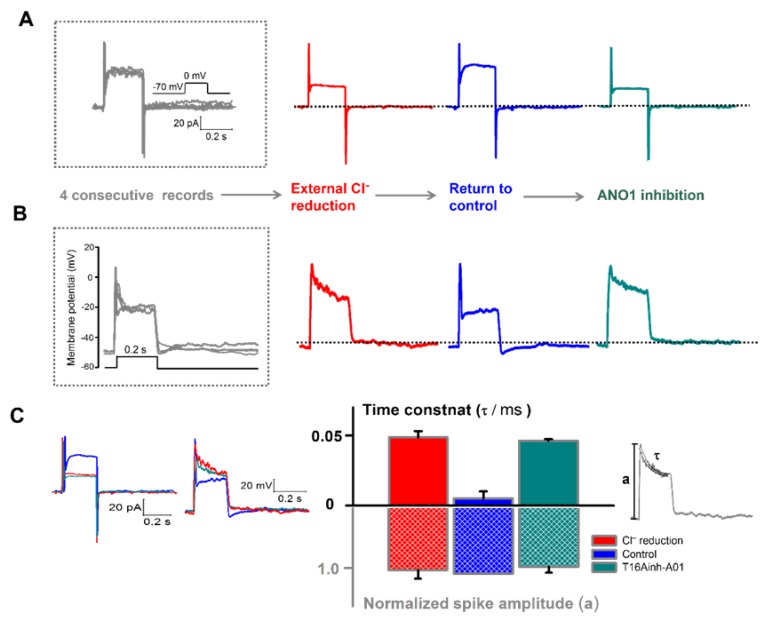
The voltage-dependent component of ANO1 is involved in the rapid decay of the initial phase of the waveform. (**A**) Four consecutive voltage-clamp recordings (10 mV intervals from −70 mV holding potential for 250 ms) were performed to determine the control condition of the bipolar cells lacking I_tail_. From the control condition (gray), the responses were detected by changing the extracellular solutions in this order: low Cl^−^, control, and with T16Ainh-A01. (**B**) After switching to the current-clamp mode, the control waveforms (gray) and cell membrane potential were detected using the same process. The cells were depolarized by current injection (≈20 pA) from a resting potential of approximately −50 mV. Representative traces are shown in red (low Cl^−^), blue (control), and dark cyan (T16Ainh-A01). (**C**) The traces obtained from the two modes are superimposed to compare the differences between the outward current component and the decay of the initial potential phase. The decay time constant τ was obtained from the single exponential fit of the initial phase of the waveform. Each amplitude spike was measured and shown in the inset. The time constant in the decay phase (top row) compared with the normalized spike amplitude (bottom row) was represented by a bar graph for each condition. The results were normalized by setting the maximum value as 1 (*n* = 10, *p* < 0.05, ANOVA, τ_red_ = 0.048 ± 0.0046 ms, τ_blu e_= 0.0051 ± 0.0049 ms, τ_dark cyan_ = 0.046 ± 0.0012 ms; a_red_ = 0.94 ± 0.13, a_blue_ = 1, a_dark cyan_ = 0.90 ± 0.89; *n* = 6, error bars represent S.E.M).

**Figure 4 cells-09-00543-f004:**
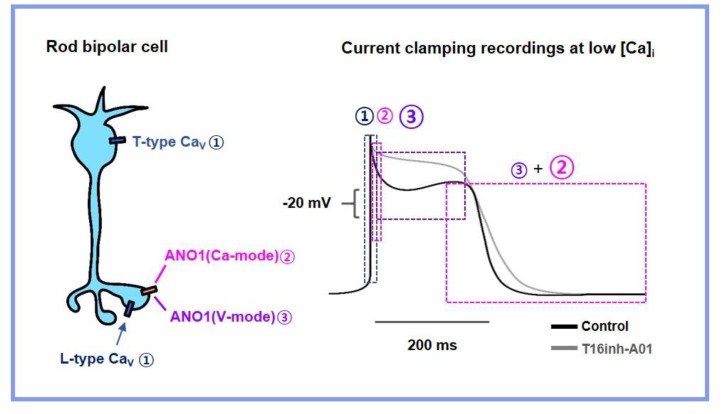
The dual gating mode of ANO1 in rod bipolar cells. (1) Voltage-gated Ca^2+^ influx acts as a trigger for the (2) activation of voltage-dependent ANO1, which contributes to the rapid repolarization of the cell membrane potential and has outward rectifying properties at voltages above −20 mV. (3) At lower voltage conditions, only the slow-running Ca^2+^ mode is ready to open, which may last longer when there is sufficient intracellular Ca^2+^.

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
