# Peer review of "Calcium- and Voltage-Dependent Dual Gating ANO1 is an Intrinsic Determinant of Repolarization in Rod Bipolar Cells of the Mouse Retina"

_cells, 2020, doi:10.3390/cells9030543_

Round 1
Reviewer 1 Report
This short paper shows evidence for voltage-dependence in activation of the calcium-activated chloride channels of rod bipolar cells and that these channels help to make depolarizing responses more transient. The experiments are technically proficient. The advance is incremental. It is well established that Ano1 channels show both calcium and voltage-dependence. Beyond showing that there is some voltage-dependence to calcium-activated chloride channels, there is little insight into the mechanisms or properties of these channels. It is also not clear to me why only some channels show this behavior. Understanding what controls the expression and activity of these channels and how they shape behavior in vivo would strengthen this study. The study could benefit from some additional English editing.
Major concerns:
Since the conclusion that they are studying Ano1 channels rather than Ano2 rests in part on the pharmacological specificity of these inhibitors., please provide more details and references concerning the specificity of T16A01 and CaCCinhA01.
Line 143. How many cells are “some”? What fraction of cells showed the apparent voltage-dependence? The text says that some cells showed continuously increasing tail currents, but the traces in Fig. 2 show no tail currents. Were tail currents consistently associated with this voltage-dependent increase in outward currents? Why do some cells show this feature and not others? Were all of the experiments done with isolated cells? Were some cells more damaged? Did some cells lack terminals? Were both types of responses seen in rod bipolar cells from slices?
Minor:
Line 23 change “period” to “periods”.
Line 45 change “have” to “has”
Line 98 change “gramicidine” to “gramicidin”
Line 120. What were the concentrations of T16Ainh-A01 and CaCCinh-A01? Was there recovery after washout?
Line 121 What was the concentration of BAPTA? How was it applied?
Line 127. References 42 and 43 should be 41 and 42.
Line 130: Change “simultaneously” to” simultaneous”
Line 185. How much EGTA was used?
Line 204. The transients were fit with two exponentials but only one time constant was reported. Which one?
Fig. 3. Rather than reporting a normalized time constant in Fig. 3, reporting the actual time constants would be more informative to the reader.
Figure S2. I don’t understand what the inset shows in Fig. S2. The explanation in the legend is not clear.
Please report the predicted value of ECl which is undoubtedly close to zero mV.
Author Response
Response to Reviewer 1
I wish to thank the reviewers for taking time in reading our manuscript and providing valuable feedback. We have incorporated your insightful suggestions in our study and have revised our manuscript. For your convenience, the reviewer’s comments, our replies, and quotations from the text are in bold, normal, and underlined fonts, respectively.
General comments: The study could benefit from some additional English editing.
Response: Per your comment, our manuscript has been edited by a native English speaker from Editage. We have attached the “Certificate for English Editing” at the end of this file for your reference.
Major concerns:
Since the conclusion that they are studying Ano1 channels rather than Ano2 rests in part on the pharmacological specificity of these inhibitors., please provide more details and references concerning the specificity of T16A01 and CaCCinhA01.
Response: We understand your concern. However, we aimed to characterize ANO1 and elucidate its role in the rod bipolar cells of the mouse retina. We neither investigated the specificity of T16A01 nor CaCCinhA01 on ANO1 and ANO2 channels. Previous studies (Stöhr et al., (2009) J Neurosci; Keckeis et al., (2017) Exp Eye Res) have reported ANO2 was expressed in photoreceptor terminals and retinal pigment epithelium but not in rod bipolar cells of the mouse retina. Therefore, we speculated that T16A01 and CaCCinhA01 can act as ANO1 inhibitors in our pharmacological experiments involving rod bipolar cells of the mouse retina.
Line 143. How many cells are “some”? What fraction of cells showed the apparent voltage-dependence?
Response: Thank you for your comment. Fifteen out of 25, or 60% of the dissociated cells, showed robust Itail, whereas in the rest of cells (10 out of 25), the amount was too small to be detected. In retinal slices, cells showing voltage-dependent Itail were also detected (n=4). We have clarified this accordingly. Please see line 148.
The text says that some cells showed continuously increasing tail currents, but the traces in Fig. 2 show no tail currents. Were tail currents consistently associated with this voltage-dependent increase in outward currents? Why do some cells show this feature and not others?
Response: The voltage-dependency of ANO1 can be inferred from Figure S1 and has been confirmed by Figure 2. Voltage-dependent Itail suggests the existence of a voltage-dependent component of the ANO1 current; thus, we intended to minimize the Itail using 5 mM EGTA to clearly isolate the said voltage-dependent component in Figure 2.
The difference between cells having Ca2+-dependent Itail and voltage-dependent Itail was the prominent inward Ca2+ current. Usually, cells with voltage-dependent Itail show less prominent Ca2+ current. Although ANO1 has both voltage and Ca2+ dependencies, the voltage-dependent Itail component could be masked by the Ca2+ dependency.
Were all of the experiments done with isolated cells?
Response: We performed the experiments using dissociated cells and retinal slices.
Were some cells more damaged? Did some cells lack terminals?
Response: We have excluded both damaged cells and cells without terminals.
Were both types of responses seen in rod bipolar cells from slices?
Response: Yes, both types of responses have been observed from slice specimen, as shown in Figure 1C and S1.
Minor:
Line 23 change “period” to “periods”.
Response: We have changed “period” to “periods”. Please see line 23.
Line 45 change “have” to “has”.
Response: We have changed “have” to “has”. Please see line 49.
Line 98 change “gramicidine” to “gramicidin”.
Response: We have changed “gramicidine” to “gramicidin”. Please see line 102.
Line 120. What were the concentrations of T16Ainh-A01 and CaCCinh-A01? Was there recovery after washout?
Response: We used 40 μM T16Ainh-A01 and 40 μM CaCCinh-A01 in the retinal slices and indicated this in the manuscript. Please see line 124. The inhibitory effects of both inhibitors were not recovered after washout. However, the drug effects were recovered in the dissociated conditions even after washout.
Line 121. What was the concentration of BAPTA? How was it applied?
Response: We added 5 mM BAPTA to the intracellular solution. Its application has been accordingly discussed in detail. Please see lines 125 and 100.
Line 127. References 42 and 43 should be 41 and 42.
Response: We appreciate your attention to detail. We have corrected this accordingly.
Line 130. Change “simultaneously” to” simultaneous”
Response: We have changed “simultaneously” to “simultaneous”. Please see line 134.
Line 185. How much EGTA was used?
Response: We used 5 mM EGTA to reduce Ca2+ dependency, and this has been added in the manuscript. Please see the Line 189.
Line 204. The transients were fit with two exponentials but only one time constant was reported. Which one?
Response: We apologize for our mistake. As you have pointed out, the transients were fit with the first-order exponential function. We have changed “the double exponential fit” to “the single exponential fit”. Please see lines 205-206 and 223.
Fig. 3. Rather than reporting a normalized time constant in Fig. 3, reporting the actual time constants would be more informative to the reader.
Response: Per your suggestion, we have modified Figure 3 and the corresponding figure legends. We provided the actual values and plotted them.
The time constant in the decay phase (top row) compared with the normalized spike amplitude (bottom row) was represented by a bar graph for each condition. The results were normalized by setting the maximum value as 1 (τred = 0.048 ± 0.0046 ms , τblue= 0.0051 ± 0.0049 ms, τdark cyan= 0.046 ± 0.0012 ms ; ared = 0.94 ± 0.13, ablue= 1, adark cyan= 0.90 ± 0.89; n=6, error bars represent S.E.M). Please see revised Figure 3 and lines 224 – 228.
Figure S2. I don’t understand what the inset shows in Fig. S2. The explanation in the legend is not clear.
Response: We apologize for the confusion. We have revised it accordingly.
Figure S2. Inhibition of the ICa abolished outward rectifying current of the ANO1 in the retinal slice samples. A series of whole cell currents (from –60 to 40 mV at 10-mV interval) was recorded to confirm the change in the outward current components after the administration of T-type and L-type VGCC inhibitors with 1 mM EGTA (inset). Compared with figure 1C, Ca2+ influx and tail were not present at–30 mV. The outward voltage-dependent traces also did not appear at 0 and 30 mV. (For each trace in pink, dark cyan, and navy, the voltage is –30 mV, 0 mV, and 30 mV, respectively). Please see lines 317 -322.
Please report the predicted value of ECl which is undoubtedly close to zero mV.
Response: We calculated ECl and obtained 0. We have added this in the Materials and Methods section. Please see lines 99-100.
Reviewer 2 Report
The paper by Paik and co-workers describes the dual-gating mode of ANO1, a calcium-activated chloride channel, in retinal rod bipolar cells, stressing that the physiological activity of ANO1 is strongly associated with voltage gated calcium channel. Furthermore, they suggest that this dual gating ‘mechanism’ modulate the excitation in these cellular type. They conclude that ANO1, acting as an intrinsic inhibitory modulator of excitation, could be a good candidate for replacing the regulation role of BK and SK channels in other neural cells excitation.
The study, as a whole, is well conducted and the results described in the manuscript are potentially interesting.
I have some suggestions that might improve further the manuscript:
From line 57 to line 64: you state that’…this study aimed to determine the gating mechanism of endogenous ANO1 in mouse retinal rod bipolar cells, which strongly express ANO1 and have ICl(Ca), as well as compare its characteristics with those of transfected ANO1 in cultured cells…’.
Actually in the results section, I didn’t find any comparison between mouse retinal bipolar cells and transfected ANO1 cultured cells data.
Line 118: Why 0.5 mM EGTA was considered ‘out of physiological condition’ whether 1 mM EGTA is considered ‘physiological’? Please, comment/explain or add a reference.
Why did the authors use such elevated concentrations of Mibefradil and Nifedipine (Fig1)? And why they decided to change these concentrations (data reported in Fig2). Is there any reference on that? Please justify/comment. By the way, at line 158-159 the concentration of Mibefradil is reported to be 10 µM while in the legend of Fig2 is written ’20 µM Mibefradil’ (line 177). Which is the exact concentration used for this set of experiments?
Mibefradil has been reported to block other channels than T type calcium channels. In order to strengthen their conclusions authors should try another T type calcium channels blocker, i.e. TTA-A2.
Nifedipine at such a high dose (40 µM) is not considered a specific L type calcium blocker (i.e. potassium channel, e.g. kv1.2, are also modulated by high concentration of Nifedipine). Please, confirm your results using, at least, a lower concentration of Nifedipine (3 µM is enough to block L type calcium channels). Moreover, it could be interesting to test for the presence, and eventually the importance, of an E4031 (specific blockers for hERG and Kv2.1-containing channels) sensitive current in Itail
Please, explain how data in fig1A right panel and in fig1D are normalised (fig 1A, 1D)
Looking at the graph in fig 2D and fig. 2E it seems that Mibefradil has not effect whereas Nifedipine still reduces the current, how do you explain that? Does L and T type calcium channel have a differential role in the regulation of ANO1?
Round 2
Reviewer 1 Report
I strongly recommend that the authors discuss, with appropriate citations, the pharmacological specificity ofT16A01 and CaCCinhA01. I appreciate that they have not themselves studied the specificity in detail, but other studies have looked at these compounds.
Author Response
Comments and suggestions: I strongly recommend that the authors discuss, with appropriate citations, the pharmacological specificity of T16A01 and CaCCinhA01. I appreciate that they have not themselves studied the specificity in detail, but other studies have looked at these compounds.
Ans.: Thank you for your suggestion. Unfortunately, we did not study the pharmacological specificity of the T16A01 and CaCCinhA01. We apologize for not appropriately replying to your 1st comment in the 1st revision. Following your suggestion, we have inserted a few sentences about the pharmacological specificity of T16A01 and CaCCinhA01 in the Discussion section with appropriate citations. Please refer to lines 237-243. “Regarding ANO1-specific inhibitors, it is worth mentioning that both CaCCinh-A01 and T16Ainh-A01 reportedly do not pharmacologically discriminate between ANO1 and ANO2 [42]. Nevertheless, CaCCinh-A01 and T16Ainh-A01 have been widely used as ANO1-specific inhibitors [41, 42, 45]. Moreover, previous immunohistochemical studies demonstrated that the mouse retinal rod bipolar cells did not express ANO2 [16, 45], while they strongly expressed ANO1 [13]. Therefore, it is reasonable that, in this study, both inhibitors were used as ANO1-specific inhibitors.”
Reviewer 2 Report
Authors answered to all my comments/concerns/question
Author Response
Thank you for your generous comments and understanding.